# Consonant and vowel articulation accuracy in younger and middle-aged Spanish healthy adults

Ignacio Moreno–Torres 1☉*, Enrique Nava 2☉

1 Department of Spanish Language, University of Málaga, Málaga, Spain, 2 Department of Communications Engineering, University of Málaga, Málaga, Spain

☉ These authors contributed equally to this work.
* imoreno@uma.es

**Data Availability Statement:** All the scripts as well as the audio corpora of experiment 1 is stored in a public repository: URL: https://github.com/Caliope-SpeechProcessingLab/ASICAKaldiGMMRecipe/tree/v1.0.1 DOI: 10.5281/zenodo.4013245. The

## Abstract

Children acquire vowels earlier than consonants, and the former are less vulnerable to speech disorders than the latter. This study explores the hypothesis that a similar contrast exists later in life and that consonants are more vulnerable to ageing than vowels. Data was obtained with two experiments comparing the speech of Younger Adults (YAs) and Middle–aged Adults (MAs). In the first experiment an Automatic Speech Recognition (ASR) system was trained with a balanced corpus of 29 YAs and 27 MAs. The productions of each speaker were obtained in a Spanish language word (W) and non–word (NW) repetition task. The performance of the system was evaluated with the same corpus used for training using a cross validation approach. The ASR system recognized to a similar extent the Ws of both groups of speakers, but it was more successful with the NWs of the YAs than with those of the MAs. Detailed error analysis revealed that the MA speakers scored below the YA speakers for consonants and also for the place and manner of articulation features; the results were almost identical in both groups of speakers for vowels and for the voicing feature. In the second experiment a group of healthy native listeners was asked to recognize isolated syllables presented with background noise. The target speakers were one YA and one MA that had taken part in the first experiment. The results were consistent with those of the ASR experiment: the manner and place of articulation were better recognized, and vowels and voicing were worse recognized, in the YA speaker than in the MA speaker. We conclude that consonant articulation is more vulnerable to ageing than vowel articulation. Future studies should explore whether or not these early and selective changes in articulation accuracy might be caused by changes in speech perception skills (e.g., in auditory temporal processing).

## Introduction

Aging is associated with multiple changes in physiology and cognitive skills supporting speech articulation. For instance, changes have been described in the stiffness of the vocal folds, in the strength and mobility of the tongue and also in the movement sequencing skills needed to

actual results obtained in the ASR and HSR experiments are included as supplementary material.

**Funding:** IMT an EN received a grant from Ministerio de Ciencia, Innovación y Universidades (RTI2018- 094846-B-I00) and a grant from Junta de Andalucia (UMA18-FEDERJA-021). Speech data acquisition was funded by Ministerio de Economía, Industria y Competitividad, Gobierno de España, PI16/01514. The funders had no role in study design, data collection and analysis, decision to publish, or preparation of the manuscript.

**Competing interests:** IMT and EN have no competing interest in this study.

generate articulation programs [1–4]. Also, there is increasing evidence that auditory processing skills tend to decay with age [5, 6], and that this may impact speech articulation skills [7]. Thus, it is not surprising that articulation accuracy decreases with aging, and that many Older Adults (OA: > 65 years old) show slow or atypical rhythm and also variable segmental errors [4, 8–14].

Understanding the precise nature of these errors might be most valuable both from a theoretical and from a clinical perspective. However, to date there is limited information regarding the precise error patterns observed in OAs. Also the fact that most studies have examined a few languages, mostly English, makes it difficult to generalize the results to less studied languages. These considerations motivated our interest in speech errors in Spanish language healthy speakers.

In order to understand the effects of ageing it seems reasonable to consider separately the suprasegmental and the segmental aspects of the speech. As to the first ones, many studies have observed that a characteristic of Younger Adults (YAs: < 35 years old) is that they produce speech more rapidly than Older Adults (OA: > 65 years old) or Middle-aged Adults (MA: 50–60 years old). It has also been observed that the errors of OAs are more frequent in specific prosodic positions (e.g., in coda position in the syllable; [3, 4, 10–12]).

Regarding segmental data, there is evidence that OAs produce more errors than YAs or MAs [8, 9, 12–14]. For instance, [9] carried out a perceptual judgment experiment of oral diadochokinetic performances of 10 healthy YA and 10 healthy OAs. Expert listeners scored the speech of the OAs systematically worse on a series of perceptual dimensions including consonant precision, vowel precision, and voice quality. Similar results were obtained by [8]. Focusing on vowels exclusively, some acoustic studies have observed diverse changes in the vowels of OAs compared to those of YAs (e.g., centralization of formant frequencies, decrease in F1 frequency, etc.; [13, 14]). However, in a study with French speakers, [12] found differences between YA and OAs in nasal vowels but not in oral ones. Also, in a recent study exploring formant frequencies in 53 adults between the ages of twenty to ninety-two years, the authors found that the formant frequencies did not change significantly [15]. Altogether these results indicate that ageing might be associated with a decrease in articulation rate and a difficulty to produce speech sounds in positions that require increased effort (e.g., in coda position). Results also suggest that the error types might vary cross-linguistically. Finally, and given that there is agreement among researchers regarding consonant accuracy but not so much vowel accuracy, it is possible that there is a contrast between the two categories, with consonants being more vulnerable than vowels. The possibility that consonants are more vulnerable than vowels would not be surprising: studies in child speech development have long noted that toddlers start producing vowels well before consonants [16]; also it has been observed that children with speech planning deficits do succeed in learning their first vowels but they struggle to learn their first consonants [17–19]. Interestingly, delays in consonant acquisition seem to vary cross-linguistically, and might be more common in languages with relatively simple syllable structures such as Hebrew and Spanish (in contrast with languages with complex syllable structures such as Dutch or English). Based on the above evidence, we hypothesized that consonant production might be more vulnerable to ageing than vowel production, though the degree to which there is a contrast in vulnerability might depend on the target language.

In order to address this question it is relevant to make some methodological considerations. Note that many studies have analysed accuracy using perceptual ratings or acoustic analyses. Those approaches might provide reliable results for single case or small group studies, but they might be impractical to explore large groups of speakers. Note also that small declines in accuracy might not have perceptual consequences, and speakers can use compensatory strategies such as reducing the rate of articulation [20]), for which minor phonetic differences may pass

undetected. One alternative approach to classical methodologies consists in using ASR systems [21]. ASR technology has been used in the past to compare the accuracy of healthy subjects with that of subjects with speech disorders [22]. Also, it has been noted that it can be useful to explore relatively small differences such as those analysed in sociolinguistics or dialectal research [22, 23]. One potential advantage of ASR systems is that they allow to analyse the acoustic information exclusively (i.e., ignoring contextual, visual, lexical. . . information) Furthermore, by creating sufficiently large speech databases it should be possible to explore phonetic trends in the social network. However, it is important to be cautious when using ASR systems to study articulation accuracy (or intelligibility). Note that ideally we would expect the ASR system to recognize all and only those speech sounds that are produced accurately (according to an ideal or expert listener). However, previous studies comparing different ASR systems and Human Speech Recognition (HSR) have shown that, despite the overall performance level similarities, the two might not always rely on the same properties of the acoustic speech waveform (see [24–26]). These considerations show that it is necessary to complement ASR data with human based analyses [27]. One such approach consists in asking naïve listeners to recognize speech sounds presented in adverse conditions (e.g., with a background noise; [28]).

Here we summarize the results of two experiments that compare the articulation accuracy of MA and YA speakers. There are various reasons that motivated our interest in MAs (instead of directly studying OAs). In the first place, while the cognitive and physiological differences between YAs are MAs are very small, we assumed that even if these differences might not be sufficiently severe to be considered clinically relevant, they might be detected by an ASR system if the database used to train the system was sufficiently large and appropriately balanced. In the second place, the fact that, as compared with OAs, MAs are a relatively homogeneous group and with very limited auditory or cognitive deficits would reduce the use of compensatory strategies (e.g., slow articulation), which might facilitate the identification of age related group differences. Thus, we expected that comparing YAs and MAs would provide valuable information to test the hypothesis that consonants are more vulnerable to ageing than vowels.

The first experiment consisted in training a standard ASR system (i.e., Kaldi [21]) with a balanced corpus of YA and MA speakers and then evaluating it with data from the same age groups. The corpus used to train the system consisted of Words (Ws) and Non-Words (NWs) produced during a repetition task. All the syllables consisted of a single consonant plus a single vowel. This task is used in our lab to evaluate MA and OA patients with speech disorders. The system used acoustic and sub-word information (i.e., phonemes and syllables) but not lexical or grammatical information. Accuracy measures (i.e., percent correct) were obtained for various phonological categories (i.e., syllables, consonants, vowels, consonant features and specific phonemes). Based on previous evidence [29]) we anticipated that the system would find differences between the two groups, particularly for consonants, which would confirm our hypothesis.

The second experiment consisted in asking a group of healthy native speakers to recognize the isolated Consonant-Vowel syllables presented with a noisy background. The target syllables were, respectively, from one YA and one MA who also participated as speakers in the ASR experiment. Note that the data from this experiment has been analysed previously [28]. In our previous analyses we observed that while the listeners recognized the two speakers to the same extent, there were some qualitative differences between the two: generally, the vowels and consonants with formant structure (e.g., nasals and approximants) were better recognized in the MA speaker, while voiceless consonants were better recognized in in the YA speakers. Here we explore to what extent there is a parallel for these two speakers between the results of the ASR experiment and the speech in noise data.

## Materials and methods

### ASR experiment: System design

The present experiment adopted most aspects of the Gaussian Mixture Model-Hidden Markov Model (GMM-HMM) architecture and configuration implemented in the Kaldi ASR toolkit [21]. The signals were segmented using a 25 ms overlapping Hamming window with a 10 ms step. Acoustic analysis was made with 13 Mel Frequency Cepstral Coefficients (MFCC) from a 26 bands filter bank as well the corresponding velocity and acceleration coefficients (i.e., computing the difference between consecutive MFCC features). In order to guarantee that the results were speaker independent and to maximize the error lists we used a cross-evaluation approach: the same process was run 56 times (one per speaker in the database); each time the system was trained with the data from 55 speakers and evaluated with the remaining speaker.

Two aspects of the design of the ASR system were especially relevant for this study: the language model and the corpus design. As to the language model, most ASR applications are designed to recognize full words and a more or less constrained grammar. This approach is very effective for many applications, but it limits the possibility of identifying specific error patterns (e.g., p > t, b > d). In this study the lexicon was composed of all the syllable types in our repetition task, and the grammar was designed to accept any sequence of syllables. We assumed that this approach would allow us to identify the phonological errors in the YA and the MA speakers. A version of the scripts used to run this experiment can be obtained from [30]. The corpus design is described in the next section.

### ASR experiment: Corpus design and speech samples annotation

The database used for this study was composed of a total of 224 recordings obtained from of 29 YA and 27 MA native Spanish speakers (i.e., each participant produced four different recordings). All the speakers were original from the region of Andalucia and none of them had a strong Andalusian accent (e.g., ceceo or seseo) or any known hearing or speech deficit. The YA speakers were students at the University of Málaga. The study was performed according to the Declaration of Helsinki and the protocol was approved by the Local Community Ethics Committee for Clinical Trials (CEUMA: Comité Ético de Experimentación de la Universidad de Málaga) and by the Spanish Medical Agency.

In order to ensure that the two groups were similar from a sociolinguistic point of view, the MA participants were recruited with the help of the YA speakers. Two thirds of the MA speakers were relatives to one YA participant (N = 18). The remaining MA speakers were recruited among administrative and academic staff of the University of Málaga (N = 9). The YA group included 12 males and 17 females, and the mean age was 21.8 years old (Std Dev.: 4.3). The MA group included 10 males and 17 females, and the mean age was 54.8 years old (Std. Dev: 3.7). Following the indications of the IRB of the University of Málaga, which required that the speakers were informed in advance, and that data were anonymized, all the participants signed an informed consent form and the recordings were anonymized with an unique code.

The four recordings from each participant were obtained while they imitated four lists of 48 utterances (i.e., to a total of 192 utterances per participant) produced originally by a female speech therapist. Each utterance was either a real W or a NW, with Ws representing 33% of the utterances. All syllables were composed of Consonant + Vowel (e.g., pa, ka). The length of the utterances in syllables, both in Ws and in NWs, was two (52% of the utterances), three (29%) or four (19%). All the utterances had the most common prosodic pattern in Spanish, with stress in the penultimate position (e.g., /PA ta/, /pa TA ka/, /la pa TA ka/).

The recordings were obtained in a quiet room at the University of Málaga. The speakers wore AKG K240 headphones. They were placed in front of a computer screen that showed a number and subsequently produced one W or NW that the participant had to imitate. Utterances were presented every 5 seconds and a pause was made after every 48 items. The productions of the speakers were recorded using the internal microphone of a Zoom H4n Pro digital recorder. Note that internal microphones may result in relatively poor quality in the recordings. However, as the long term aim of this study is to evaluate patients in a clinical context, we decided to use technical conditions that are easily available to speech therapists. Also, based on previous experience we assumed that the impact on ASR scores of using an internal microphone instead of an external one would be relatively small.

The recordings were transcribed phonologically in two stages. In a first stage one phonetician compared the actual productions of the speakers with the target W or NW. Whenever there was an error, the phonetician annotated the actual production, otherwise the original target was used as transcription. In a second stage the system was trained and evaluated with the full database; then a second phonetician revised only the items for which the system had failed. In case of disagreement between the first and the second phonetician, the final decision was adopted by a third phonetician (i.e., the first author).

## Speech in noise experiment

The data summarized here was computed from a database obtained as part of a previous experiment exploring consonant resistance to noise [28]. In that experiment the target utterances were isolated Consonant-Vowel (CV) syllables (N = 80) produced twice by two male talkers that also took part in the ASR experiment described in this study. The two speakers were, respectively, one YA and one MA, and were identified as YA402 and MA001. The target syllables were presented with a background babble noise created by combining eight talker voices (4 female, 4 male). The individual intensity levels for the babble noise and target-CVs were adjusted according to the global root mean square power of the original sounds to be mixed, at three Signal-to-Noise Ratios (SNRs): -6 dB, -2 dB, and +2 dB. In total this resulted in 960 different stimuli (2 talkers x 2 repetitions x 80 CV x 3 SNRs). Each participant was tested only on one of the lists (i.e., 240 tokens: 1 talker x 1 repetitions x 80 CV x 3 SNRs).

Seventy-eight native Spanish speakers (41 female) participated in the experiment. All the participants were aged between 18 and 34 years and had no history of hearing loss or language disorders. The listening test was automated using a Praat MFC Experiment code with graphic user interface [31]. The listener was seated in a sound booth in front of a computer monitor and heard the stimuli via headphones (AKG K141-MKII). The computer running the Praat code was placed outside a sound-treated booth to minimize ambient noise. The monitor screen showed 86 buttons. Eighty buttons were labelled with the 80 CVs; five buttons were labelled with the five vowels (V). Finally, there was one empty Noise Only button. Every new stimulus was presented exactly one second after the listener had made his or her previous decision. Every 24 stimuli the listener was given the opportunity to take a pause. The experiment lasted on average 18 min.

## Data analysis

For the ASR experiment, the results of the 56 tests were combined to compute, for each group (i.e., YA and MA) the ratios of correct: 1) syllables, consonants and vowels; and 2) place of articulation, manner of articulation and voicing consonant features. Measures were obtained separately for Ws and NWs and also for different syllable types and utterance lengths. Similarly, for the speech-in-noise experiment, the results of the 78 judges were used to compute,

**Table 1. List of Spanish consonants.**

| Features | Values | Members |
|---|---|---|
| Manner | Plosive | p,t,k,b,d,g |
| | Affricate | ʧ |
| | Fricative | f,θ,s,ʝ,x |
| | Nasal | m,n,ɲ |
| | Approximant | l,ɾ,r |
| Place | Labial | p,b,f,m |
| | Coronal | θ,t,d,s,n,ɲ,l,r,ɾ,ʝ,ʧ |
| | Dorsal | k,g,x |
| Voicing | Voiced | b,d,g,ʝ,m,n,ɲ,l,r,ɾ |
| | Unvoiced | p,t,k,ʧ,f,θ,s,x |

for the full database and separately for the three SNR levels, the ratios of correct syllables, consonants, vowels and consonant features. For group comparison, it is important that in many cases the individual speakers, especially the YAs, might show a ceiling effect and the group distribution would not meet the conditions to use parametric tests. For this reason we used U Mann-Whitney tests for group comparisons and Spearman correlation non-parametric tests. All the statistical analyses were made using SPSS 24. In order to facilitate the interpretation of the results, data is presented graphically using histograms for the YA and MA groups. Table 1 shows the consonant inventory of standard European Spanish and shows the lists of consonants for each feature value. Note that the consonants /ɲ, ɾ/ do not appear in the speech-in-noise data because in Spanish language these phonemes do not occur in word initial position.

## Results

The results are organized as follows. In the first place, we analyze the errors identified by the expert phoneticians as well as the utterance durations. This analysis will provide a preliminary overview of the two groups of speakers. Next, we will examine the results of the ASR experiment. Finally, we present the results of the speech in noise database.

### Manual analyses of the Ws and NWs corpus

The training and testing corpus consisted of 10 452 utterances, each utterance corresponding to either a W or a NW. A total of 993 utterances (9.6%) had one or more phoneme errors according to the phoneticians. As errors resulted in new or infrequent syllable types which might not be learnt by the ASR system, these utterances were excluded from the database. Thus, the final database had 9350 utterances with 26 990 syllables (and the same number of consonants and vowels). The number of syllables in Ws and NWs was, respectively, 9 894 and 17 096. The large majority of the errors occurred in NWs (> 98%) and involved mostly consonants (> 95%).

As these manually annotated errors in NWs with might help to clarify the differences between the two groups of speakers, we provide further details about them. Consonant errors in NWs were more frequent in females (3.5%) than in males (2.7%) but the difference was not significant. The errors were between two and three times more frequent in the MA group (4.8%) than in YA group (2.0%), and the difference was statistically significant (Mann-Whitney U = 557,5, s < .005). Fig 1a shows the histogram for the two groups. Note that over 60% of the YA speakers but only a 20% of the MA speakers produced less than 2.5% consonant errors.

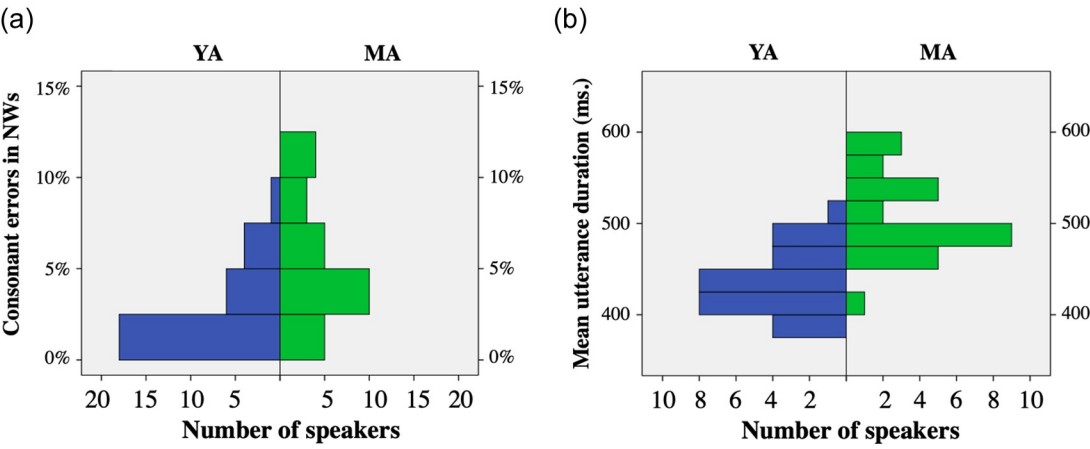

**Fig 1.** Corpus descriptors: ratio of consonant errors annotated by phoneticians (a) and utterance duration (b).

The majority of the consonant errors ($>$ 80%) involved the place of articulation (e.g., b $>$ d, t $>$ k, n $>$m). Also, there was a small number of utterances (N = 4) which were misproduced relatively frequently ($>$ 25% of the speakers).

The error patterns were similar in the two groups of speakers (i.e., place of articulation), but they were more frequent and severe (i.e., two or more errors in one utterance) in the MA than in the YA speakers. Table 2 shows some illustrative examples of these speakers' errors.

The mean utterance duration was 0.44 s. (Std. Dev. = 0.037) in the YA speakers and 0.50 s. (Std. Dev. = 0.046) in the MA speakers. The difference was statistically significant (Mann Whitney U = 700,500, s $<$ .001). As Fig 1b shows, the two groups differ clearly in this aspect. For instance, a total of 20 YA speakers, but only one MA speaker, had a mean duration equal or below .45 seconds.

## ASR of the Ws and NWs database: Preliminary results

In order to ensure that the ASR system was not biased due to the different number of male and female speakers we computed the results for these two groups separately. The female speakers were recognized somewhat better than the male speakers both in Ws (93% vs. 92%) and in NWs (92.1% vs. 91.6%), but the difference was not significant in any case. When considering

**Table 2. Sample errors in the speakers' database.**

| Target | Actual | Num errs. | YA | MA |
|---|---|---|---|---|
| di 'ne so | di 'ne so | 0 | 82% | 44% |
|  | bi 'ne so | 1 | 18% | 26% |
|  | di 'me so | 1 |  | 21% |
|  | bi 'me so | 2 |  | 21% |
| fe 'bu xo | fe 'bu xo | 0 | 72% | 45% |
|  | θe 'bu xo | 1 | 28% | 50% |
|  | θe 'bu fo | 2 |  | 4% |

Target and actual productions are transcribed using AFI symbols. The numbers indicate the percentage of speakers producing the corresponding variant.

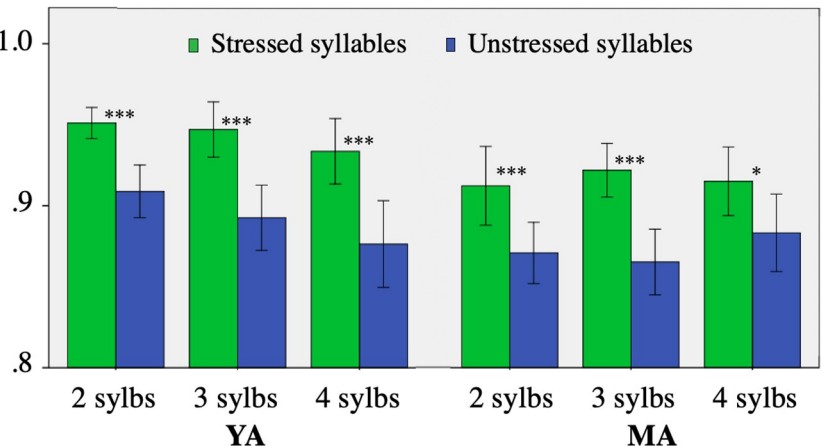

**Fig 2. Correctly recognized syllables in stressed and unstressed syllables in NWs** (***: s < .001; *: s < .05).

each age group separately the results were almost identical (i.e., the female speakers scored 1% above the male speakers). Thus, the data of male and female speakers were collapsed for subsequent analyses.

Next, we compared the percentages of correct syllables in stressed and post-stressed position and for different utterance lengths (Fig 2). The ASR system recognized stressed syllables better than post-stressed ones; note also that while in the YAs the scores tended to decrease with the length of the utterance, in the MAs the results were stable. This suggests that the MAs may have increased their articulatory effort in these long utterances (i.e., as a compensatory strategy). However, as the effect was small and only in one syllable of a group of utterances, we assumed that it would have no effect on the general scores.

Next we computed the scores for syllable in Ws and NWs separately. Fig 3 shows the ratio of syllables correctly recognized in Ws and in NWs, and both in the YA and the MA speakers. The utterances of the YA speakers were better recognized than the utterances of the MA speakers both in Ws and NWs. However, the difference was significant only in the case of NWs (Mann-Whitney U = 222.000, s = 0.005). This shows that the group difference was relatively small, but it increased in the more demanding condition (i.e., with NWs). Accordingly, in the rest of this section we will only present the results obtained in NWs.

Finally, we computed the Spearman correlations between the rate of articulation, the phonological errors annotated by the phoneticians and the ratios of syllables recognized automatically. The correlations were not significant in any case. A close inspection of the individual data confirmed that there were both well recognized (> +1 std. dev) and poorly recognized (< − 1 std. dev) speakers that articulated rapidly and slowly. Thus, it suggests that the rate of articulation is independent from the accuracy of articulation.

## ASR experiment: Vowels, consonants, vowels and consonant features

The ASR system recognized vowels to a similar extent in the YA and MA groups (97.9% vs 97.5%). In contrast, the YA speakers scored above the MA speakers for consonants (94.9% vs. 92.5%), and the difference was significant (Mann-Whitney U = 204.000; s = 0.02). The results for the three consonant features revealed a contrast between place and manner of articulation, on the one hand, and voicing, on the other. In the case of place and manner of articulation the YAs scored above the MAs, and the difference was significant both for the place of articulation

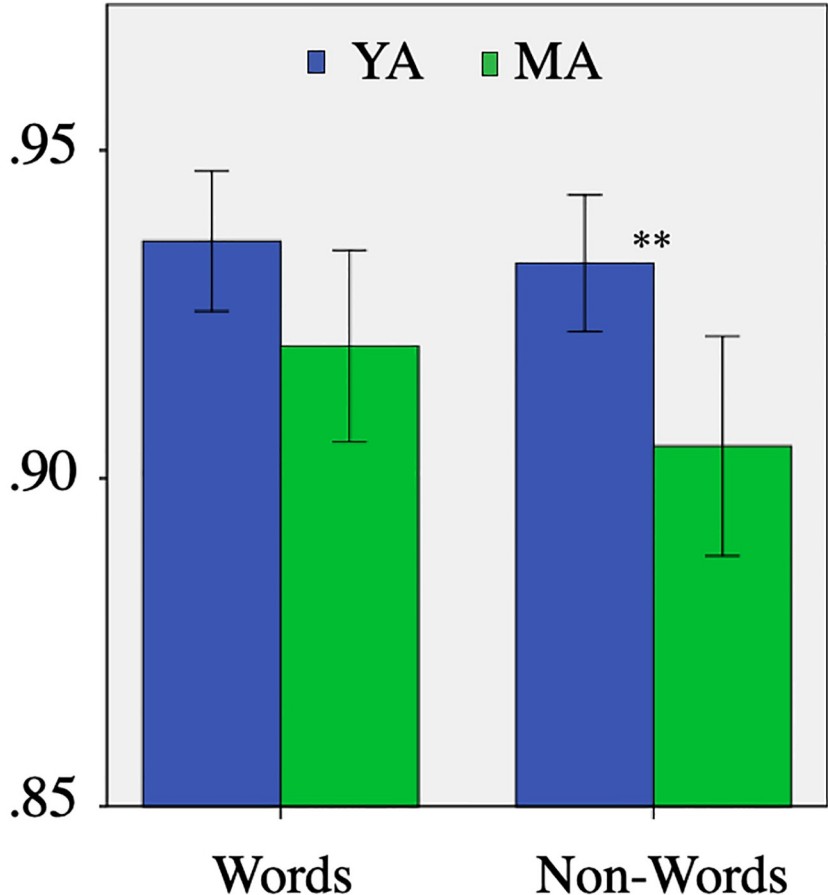

**Fig 3. Correctly recognized syllables in Ws and NWs (\*\*: s < .01).**

(Mann-Whitney U = 234.000; s = 0.010)) and for the manner of articulation (Mann-Whitney U = 271.500; s = 0.049). As for voicing, the scores were almost identical in the two groups.

In order to further understand the group differences Fig 4 presents the histograms for the two groups of speakers. Note that in the case of vowels and the voicing feature the histograms indicate that the two groups are almost identical. In contrast, in the other three cases, there is an decrease in the number of high scores and an increase in the number of low scores in the MA group compared with the YA group.

In order to clarify whether the high scores with vowels were due to compensatory strategies, we repeated the analyses presented in Fig 1 independently for vowels and for consonants. The results revealed that the trends for consonants and for vowels were the same as in the case of syllables: the ratios of correctly recognized tokens were higher in stressed than in unstressed syllables; and the ratio of ratio of correctly recognized consonants and vowels tended to decrease with the utterance length in the YA group but not in the MA group. Thus, the results indicated that the MA speakers increased the articulatory effort both in vowels and in consonants (i.e., there was not vowel specific compensation).

Next, in order to clarify the relationship between the results of the ASR system and the errors annotated by the phoneticians we computed the Spearman correlation between several ASR measures and the ratio of consonant errors annotated manually. The same analyses were

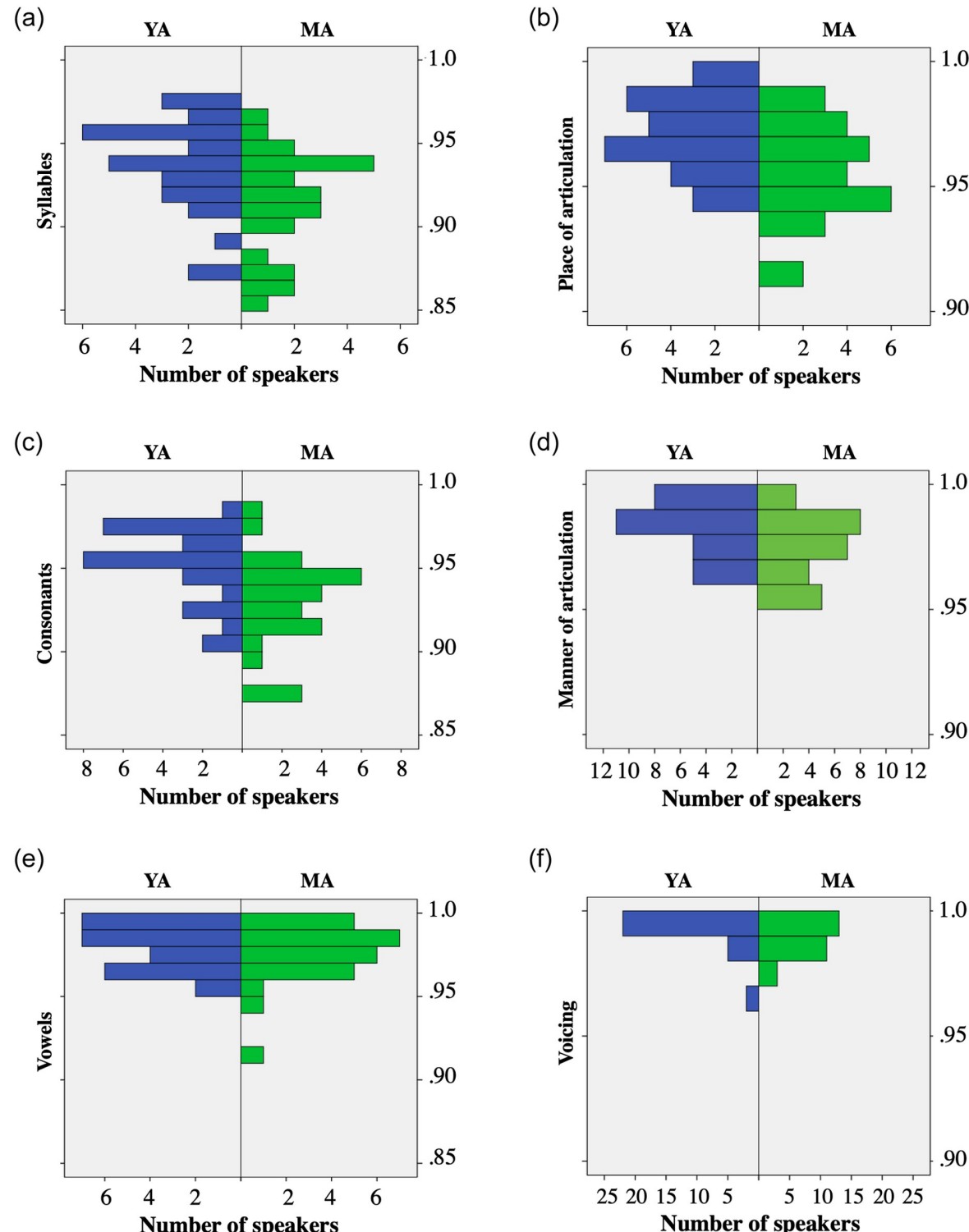

**Fig 4.** Group results in NWs: syllables, consonants and vowels (left) and consonant features (right).

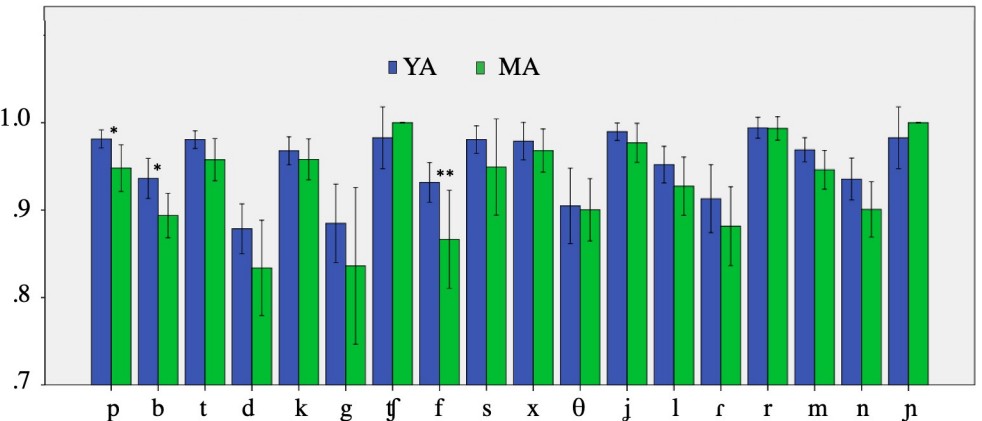

**Fig 5. Group results for 18 consonant types.**

carried for the full set of speakers and for age and sex subgroups. When the full list of speakers was included in the analyses, the ratios of errors annotated by the phoneticians were not correlated with any of the ASR measures. The same results were obtained when analysing the two age groups separately and also for the male participants. However, in the case of the female participants there was a weak but significant correlation between the consonants and place of articulation errors annotated by the human experts and the corresponding measures computed from the ASR results. For consonants: Spearman r = .39; s = .021. For the place of articulation feature: r = .34; s = .047). This indicates that in the case of the female participants there might be a link between the two measures.

Finally, we compared the scores for each consonant type separately (see Fig 5). In most cases the mean scores of the YA speakers were above those of the MA speakers. However, the group difference was significant only in three cases, all of which are labial consonants (i.e., /b, p, f/).

## Speech-in-noise experiment

In order to facilitate the interpretation of the speech-in-noise experiment, Table 3 summarizes the results of speakers YA402 and MA001 with NWs in the ASR experiment; the table also includes the full group (MA + YA) mean and standard deviation. With the exception of the voicing feature, YA402 scored clearly above the group mean. In contrast, MA001 scored below the group mean for all the measures except for vowels and the voicing feature. This means that

**Table 3. Individual and group results in the ASR experiment.**

|  | YA402 | MA001 | Group Mean | Group Std Dev |
|---|---|---|---|---|
| Mean duration | 0.51 s. | 0.58 s. | 0.49 s. | 0.05 s. |
| Correct syllables | 95% | 84% | 92% | 3% |
| Correct consonants | 96% | 86% | 94% | 3% |
| Correct vowels | 99% | 97% | 98% | 3% |
| Correct place | 99% | 91% | 96% | 2% |
| Correct manner | 99% | 94% | 98% | 1% |
| Correct voicing | 97% | 98% | 99% | 1% |

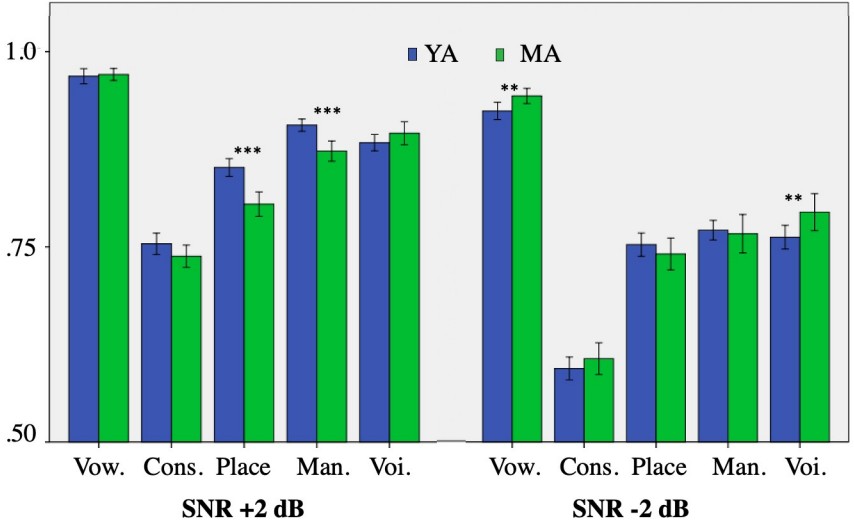

**Fig 6. Results in the speech-in-noise task.** Vowels, consonants and features (Mann-Whitney U *** s < .001; ** s < .01).

these two speakers are well differentiated in terms of how the ASR system recognizes them. Also, they can be considered as representative of their respective age groups.

As a first step we computed the ratios of phoneme and feature recognition with the full database (i.e., by collapsing the data of the three SNR levels). The results were almost in in the two speakers for syllables (55%) and for consonants (56%). For vowels YA402 scored below MA001 (87% versus 89%) and the difference was statistically significant (Mann Whitney U = 404.000, s = 0.005). Finally, for the consonant features, YA402 scored above MA001 for manner (73% versus 72%), identical for place of articulation (70% versus 70%), and below for voicing (73% versus 75%), but the difference was not significant in any case.

Next, as the SNR level might impact differently vowels, consonants and features, we repeated the same analyses separately for the three SNR levels (see Fig 6). With a SNR of +2 dB (i.e., relatively little noise), the place and manner of articulation were better recognized in YA402 than in MA001, and the difference was statistically significant. In contrast, the vowels, consonants and the voicing feature were recognized to a similar extent in both speakers. With a SNR of -2 dB, vowels and the voicing feature of MA001 were better recognized than those of YA402; in contrast the results for consonants as well as for place and manner of articulation were similar for the two speakers. With a SNR of -6 dB the results were almost identical to those of -2 dB. As for the results for specific consonants, half of the consonants were better recognized in one of the speakers than in the other speaker. The consonants that were better recognized in YA402 were the voiceless ones, while the consonants that were better recognized in MA001 were the voiced ones. Thus, the results for specific consonants do not match the results obtained with the ASR.

## Discussion

The main aim of this study was to explore whether there are relevant differences in the articulation accuracy of YA and MA speakers. For that end we analysed to what extent an ASR system, trained with a balanced corpus of YA and MA speakers, was equally effective in

recognizing the two groups of speakers. Also, in order to ensure that the results of the ASR system could be considered a good approximation to articulation accuracy, we compared the results of the ASR system with the results of a speech in a noise recognition task. We will analyse in the first place to what extent the results of the ASR and the speech-in-noise experiments are comparable and reliable. Then we will discuss the theoretical and clinical implications of the ASR experiment.

## ASR as model of Human Speech Recognition

As noted in the introduction, it is unclear to what extent ASR systems such as the one used here access the same acoustic information in the speech signal as humans do. A detailed analysis of this issue is out of the scope of this study. However, given that the results indicate that the place of articulation feature provides a key contrast between the YA and the MA speakers, and given that this feature was the worst recognized one by the ASR system, we will briefly consider the results for this feature. Note that primary cues to the place of articulation are formant transitions [32], and that recognizing formant transitions requires access to fine temporal fine information (see [33]). As in this study we used a 25 ms window, it is possible that the ASR recognized poorly the formant transitions. This means that a decrease in the rate of recognition as observed in the results might not necessarily imply that a decline in articulation skills does take place. However, there are reasons to consider that the results of this study, at least as regards the differences between YA and MA speakers, are not merely a statistical artifact.

Note that the place of articulation errors were the most common ones observed during the transcription process, and that these errors were more frequent in the MA group than in the YA group. Two possible causes may explain these errors. One is that the MA group had minor auditory temporal processing deficits [5, 34], which may lead to poor recognition of the place of articulation [33] and which, given the close link between perception and articulation, may negatively impact articulation accuracy selectively for this feature [7]. Alternatively, it might occur that the errors observed by the human experts were due to a decline in cognitive skills supporting articulation (e.g., motor sequencing— [12]). Thus, independently of the underlying cause, the articulation skills of the MA group might be poorer than those of the YA group, which agrees with the data obtained with the ASR system.

A different approach to determine the reliability of the ASR system consists in considering to what extent the system scores were sensitive to the known variability in speech accuracy. For this end, it seems relevant to consider within–speaker differences for Ws and NWs, within–utterance prosodic positions differences and also differences for the utterance length in syllables. The ASR scores were higher in Ws than in NWs, and also in stressed than in post-stressed syllables. As for the syllable length, the scores decreased slightly in the case of YAs but not in the case of MAs. This last result suggests that the MAs may have increased their articulatory effort in this last case (i.e., as a compensatory strategy). Thus, altogether the results indicate that the ASR system is highly sensitive to minor variations in speech accuracy, which further reinforces its interest to measure accuracy changes with age.

A more stringent approach to analyse the reliability of the ASR system consists in comparing its results with those obtained in a HSR task as the one described above. Globally the HSR results show that the manner and place of articulation features were easier to recognize in the YA speaker than in the MA speaker, while vowels and also the voicing feature were better recognized in the MA speaker than in the YA speaker. Interestingly, the robustness varied as a function of the SNR levels. This is possibly related to the fact that different phonological contrasts are associated with acoustic cues which differ in the degree of resistance to noise. In the case of place and manner of articulation, it seems that the acoustic cues are relatively

vulnerable, for which small amounts of noise may easily blur them. This is certainly the case of dynamic spectral cues to place of articulation. As for manner of articulation, it might be related to the difficulty to differentiate pairs such as fricative/stop (e.g., p/f, t/θ, k/x) due to the limited energy characteristic of these fricative phonemes. In contrast, for vowels the difference between the speakers was observable only with larger amounts of noise (i.e., SNR -2 dB). This result is probably related to the fact that vowels are characterized by spectral areas with large amounts of energy (i.e., formants) which are highly resistant to noise.

Finally, the fact that voicing seems to be independent of the other two consonant features may be related to the phonological and phonetic characteristics of this feature in the Spanish language. Note that studies in other languages have provided evidence of frequent voicing errors in noise [35, 36]. However, when the voiced/unvoiced consonant pairs which are commonly confused in those languages (in onset position) are examined in detail we observe that, for each pair, both or at least one of the consonants do not exist in Spanish. This may partly explain that errors are less common in Spanish. Indeed, in Spanish there are only three minimal pairs of consonants that differ exclusively on voicing (i.e., p/b, t/d, k/g). Furthermore, [37] has consistently argued that the voicing contrast in these three pairs can be phonetically described as one tense/lax, and that it is not the presence or absence of periodicity that serves to recognize it; rather, voicing might be cued by increased tension, which may impact F0, F1 and the total energy of the vowel. This means that (phonological) voicing recognition might depend on acoustic cues that are part of the vowel. Thus, as vowels are more resistant in the MA speaker than in the YA speaker, so might be the voicing feature. Finally, the same explanation is valid for the ASR and for the HSR results. Altogether, this further supports the view that the scores of the ASR system may provide a reasonably good measure of the articulation accuracy of YA and MA speakers.

However, and in contrast with all the above-mentioned results, the scores for the individual consonants in the HSR task did not show any clear relationship with the ASR data. In the ASR experiment, the consonants that were the least intelligible were three frontal consonants (e.g., /f, p, b/). This result would suggest a connection between the lips articulator (or visibility), on the one hand, and the degree of accuracy, on the other; note that this possibility was recently suggested in [12]. However, such results were not confirmed in the HSR task; in this case, half of the consonants were better recognized in the MA speaker (i.e., voiced consonants) and the other half were better recognized in the YA speaker (i.e., voiceless consonants). There are two explanations for this phenomenon: one is that while the ASR system and humans might be similar in terms of recognizing gross categories (e.g., consonants versus vowels); the finer the categories the larger might be the differences between the two; it is also possible that the differences are related to the presence of noise in one condition exclusively.

To conclude, the analysis of the results shows that the data obtained with the ASR experiment served to compute reliable measures of articulation accuracy for gross categories such as consonants and vowels, and also consonant features. However, it is unclear to what extent the measures obtained for specific consonants provide a reliable approximation to the accuracy observed in humans.

## Accuracy and aging

Aging is a slow and long process that has measurable consequences in many aspects, including the physiological systems and the cognitive skills that support speech production. Despite many of these changes being obvious to the naïve observer, there is limited evidence regarding the possibility that such changes might have any impact on speech accuracy. This issue is particularly relevant from a clinical perspective: if there is a natural decline in accuracy, data from

patients should be interpreted by considering the accuracy of their age groups. The results of the present study provide robust evidence that, even if the decline is very small, it does take place, and it is selective.

One issue that requires some consideration is whether the observed results might be related to phonological skills or, alternatively, with minor physiological changes. There are some indications that the observed results are not merely related to physiological changes. Firstly the distances between the YA and the MA groups were more pronounced in NWs (i.e., when the demand of cognitive resources is the highest) than in Ws. Secondly, the transcribers found more phonological errors in the MA than in the YA speakers. While some of these errors might be unrelated to speech production (e.g., they might be caused by minor hearing or attention problems), it is also possible that some of these errors are caused by minor decline in phonological processing skills. Finally, the possibility that the difference between YA and MA speakers reflects underlying cognitive changes is compatible with previous evidence that motor patterns sequencing, a skill most relevant for speech production, declines early in adulthood [10]. Thus, we conclude that our results reflect a very small but significant phonological decline and not only a physiological one.

Another issue that should be considered is the possibility that the results are produced by compensatory strategies that may have been more effective for vowels than for consonants, a possibility suggested recently by [15]. Indeed, in the corpus used for this study, the MA speakers may have shown some compensatory effects by over-articulating the final syllable in four-syllable utterances. However, as the effect was identical in consonants and vowels it is not possible to conclude that the reason why vowels are more accurate is because there are compensatory effects. Thus we conclude that consonants are more vulnerable than vowels, and also that the place and manner of articulation features are more vulnerable than voicing feature.

Based on these results it is important to consider the possible causes for these selective decays. One possible explanation for this contrast between vowels and consonants is that, as one set (i.e., vowels) has fewer members than the other (i.e., consonants) it is easier both for humans and machines to learn the former than the latter. The same occurs with the three features, as voicing has fewer values than the place or manner of articulation. While this might partly explain the results, it is important to consider other factors.

Another factor that might explain these results are the differences in required articulatory effort. It seems relevant to consider the results of child development and clinical studies. Developmental studies have long noted that toddlers start very rapidly to produce vowels (e.g., as soon as three months after birth) while consonants start to appear some months later [16]. This developmental pattern has been associated with the increased cognitive requirements involved in consonant articulation. Evidence of a dissociation between consonants and vowels has also been noted in research describing children' speech disorders. For instance, some children with impaired motor control skills produce their first words using exclusively vowels (i.e., consonant-free-words) [17–19]. Again, this has been explained as a consequence of the increased cognitive demands placed by consonant production. It seems reasonable that the same explanation can be applied in the case of aging. That is, consonant accuracy might decline earlier than vowel accuracy because the former place more cognitive demands on the speakers than the latter.

Finally, it seems relevant to consider to what extent auditory perception may contribute to these results. Note that recent neurolinguistic models of speech production have shown that the speech production system includes auditory and somatosensory feedback mechanisms that are used to control articulation accuracy [7]. It is also relevant that there are age-related changes in speech perception (i.e., peripheral high-frequency hearing loss and central deficits of auditory temporal processing; see [5, 34]). And also that the recognition of the place of

articulation feature requires efficient temporal auditory processing skills. Thus it is possible that minor deficits in auditory temporal processing result in impoverished feedback and, in the end, contribute to a reduction of the accuracy in place of articulation feature. Note that this link (though not the casual relationship) was confirmed only in the case of the female participants, a result that is compatible with the evidence that that there are sex-related differences in the development and decay of these skills [5].

## Conclusions

Given these results, it seems necessary to consider to what extent they are clinically relevant. It is important to emphasize that the fact that MA speakers are less accurate than YA speakers might have a very marginal impact on speech intelligibility and on the ability to interact with others. This is so for various reasons: 1) the differences in articulation accuracy between YAs and MAs were very small and mainly in NWs (which are not used in everyday communication); 2) the differences were observed using degraded speech or with an ASR system (which may lose part of the acoustic cues); and, finally, 3) intelligibility depends on many other factors apart from articulation accuracy (e.g., lexicon, grammar, register and style, etc.) Accordingly, the results of the present study should not be interpreted as evidence that speech articulation skills show clinically, or even linguistically, significant decline in healthy MA speakers. Rather, the observed decline could be described as statistically significant but functionally non-relevant.

However, our results provide information that might be valuable from a clinical perspective. In the first place, the fact that consonants are more vulnerable to ageing than vowels means that some of the errors observed in OA patients might be due to their age and not to any underlying speech deficit. Future studies should obtain further data about consonant articulation in OAs, as this information would be most helpful to interpret the data from OAs with acquired speech disorders. In the second place, the fact that articulation decay seems to be selective, with a larger impact on the place and manner of articulation, raises some questions as to why these specific error types were found. It might be fruitful to explore to what extent there might be a link between selective perceptual deficits (e.g., peripheral high-frequency hearing loss and central deficits of auditory temporal processing) and selective articulation decay. Clarifying these associations might be most useful in the clinical context (e.g., it might be valuable to detect the presence of temporal processing deficits, which may easily pass undetected [5]).

Future studies should explore, possibly using the same methodology as in the present study, the association between known vulnerabilities in auditory processing and specific phonological structures. For instance, it might be relevant to analyse the production of vowel monophthongs, as in /pe/ or /pi/, and vowel diphthongs, as in /pie/ or /pei/. Another aspect that should be examined is the interaction between aging and sociolinguistic or dialectal variability (e.g. for register, dialect or even the psychological state). Here, we used a balanced corpus with limited linguistic variation, which has been useful to answer the specific research questions addressed in this study. It remains to clarify whether or not the same results are obtained with more diverse groups of speakers.

From a different perspective, our results further confirm the potential interest of ASR tools to evaluate articulation accuracy. It might be fruitful to use these systems to obtain normative data from different social groups (e.g., for age, dialect, etc.). This type of information might then be most useful to evaluate the speech accuracy of patients with diverse speech disorders. Finally, our results indicate ASR systems might be used for cross-linguistic research: this type of study might be valuable to understand the precise effects of ageing in different language groups, which might be most helpful to understand how speech declines with age.

## Supporting information

**S1 Dataset. Dataset generated in the ASR experiment.** SPSS file with the full list of responses of the ASR system. Each line in the file describes one syllable token. Diverse columns categorize the type of consonant, vowel and, when present, error type.
(SAV)

**S2 Dataset. Dataset generated in the HSR experiment.** SPSS file with the full list of responses of the speech in noise experiment. Each line in the file describes one syllable token. Diverse columns categorize the type of consonant, vowel and, when present, error type.
(SAV)

**S3 Dataset.**
(ZIP)

**S1 File.**
(COLLECTION)

**S2 File.**
(DOCX)

**S3 File.**
(DOCX)

## Acknowledgments

The authors thank S. Florido Llorens for his valuable work in developing the scripts used for the ASR experiment.

## Author Contributions

**Conceptualization:** Ignacio Moreno–Torres, Enrique Nava.

**Data curation:** Ignacio Moreno–Torres.

**Formal analysis:** Ignacio Moreno–Torres, Enrique Nava.

**Funding acquisition:** Ignacio Moreno–Torres.

**Investigation:** Ignacio Moreno–Torres, Enrique Nava.

**Methodology:** Ignacio Moreno–Torres, Enrique Nava.

**Software:** Ignacio Moreno–Torres, Enrique Nava.

**Supervision:** Ignacio Moreno–Torres, Enrique Nava.

**Writing – original draft:** Ignacio Moreno–Torres.

**Writing – review & editing:** Ignacio Moreno–Torres.

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
