## [Decision Letter · Decision Letter 0]

10 Aug 2020

PONE-D-20-18733

Consonant and vowel articulation accuracy in younger and middle-aged Spanish healthy adults

PLOS ONE

Dear Dr. Moreno Torres,

Thank you for submitting your manuscript to PLOS ONE. After careful consideration, we feel that it has merit but does not fully meet PLOS ONE’s publication criteria as it currently stands. Therefore, we invite you to submit a revised version of the manuscript that addresses the points raised during the review process.

Both reviewers are enthusiastic about your work, but at the same time identified some minor weaknesses that require improvement. Please see details listed below. 

We look forward to receiving your revised manuscript.

Kind regards,

Anthony Pak-Hin Kong, Ph.D.

Academic Editor

PLOS ONE

Additional Editor Comments:

Dear Dr. Moreno Torres, 

I am writing regarding the review of manuscript# PONE-D-20-18733, "Consonant and vowel articulation accuracy in younger and middle-aged Spanish healthy adults" submitted to PLOS ONE. Two Expert Reviewers and I have reviewed your manuscript.

Both reviewers are enthusiastic about your work, but at the same time identified some minor weaknesses that require improvement. For example, Reviewer 1 would like a more extensive and comprehensive literature review on stop voicing perception in the multilingual context (instead of only focusing on the Spanish language). While you have attempted to interpret your results in various ways, Reviewer 1 and I felt that a more systematic explanation based on (and cross-referencing) other existing reported studies is needed; this also echoes Reviewer 2’s comment that observations of “consonant utterances suffering more than vowels” should be justified in terms of the associated physiological mechanism of speech production. With these changes, you may better highlight the clinical implications of your study. Finally, Reviewer 2 also suggested the analysis of signal features utilized by the ASR and study the changes in them.

I believe the feedback offered from this review can help your team to revise and resubmit the manuscript for further consideration.  Please use the enclosed comments to guide your revisions, and do not forget to include a detailed response letter indicating how each detailed comment offered here is properly addressed.

Thank you for the opportunity to review your work. I look forward to receiving the next draft of your manuscript.

  Sincerely, 

Dr. Anthony Kong

 Academic Editor, PLOS ONE

Journal Requirements:

2. We noted in your submission details that a portion of your manuscript may have been presented or published elsewhere.

"A previous analysis of the data from the speech-in-noise experiment described in this study has been published in the Journal of the Acoustical Society of America in 2017, as noted in the main the main manuscript. However, the analysis on that paper and the present one are substantially different, as are the motivations of the two studies. In the present study we inquire to what extent the results in one ASR experiment are in agreement with the speech-in-noise data. "

Please clarify whether this  publication was peer-reviewed and formally published. If this work was previously peer-reviewed and published, in the cover letter please provide the reason that this work does not constitute dual publication and should be included in the current manuscript.

Reviewers' comments:

Reviewer's Responses to Questions

**Comments to the Author**

1. Is the manuscript technically sound, and do the data support the conclusions?

Reviewer #1: Yes

Reviewer #2: Partly

2. Has the statistical analysis been performed appropriately and rigorously? 

Reviewer #1: Yes

Reviewer #2: Yes

3. Have the authors made all data underlying the findings in their manuscript fully available?

Reviewer #1: Yes

Reviewer #2: Yes

4. Is the manuscript presented in an intelligible fashion and written in standard English?

Reviewer #1: Yes

Reviewer #2: Yes

5. Review Comments to the Author

Reviewer #1: Consonant and vowel articulation accuracy in younger and middle-aged Spanish healthy adults

This study explores a hypothesis that consonants are more vulnerable to aging than vowels. The authors used an ASR system for Spanish word (W) and non--word (NW) repetitions. They found higher accuracy with the NWs of younger adults than with older people, and similarly when asked to recognize isolated syllables in background noise.

I applaud the authors for attempting to interpret their results in various ways, but there is too much speculation here, with insufficient evidence to support many potential explanations. Also, the authors seem unaware of the large literature on stop voicing perception, citing only one reference 29. Yes, Spanish is different from other languages, but stop voicing perception occurs in many languages.

Using “Preview” on the PDF file, all of the Figures’ content was located at the end of the file (annoying).

Particular changes needed:

…a group of healthy native listeners were asked …

-> …a group of healthy native listeners was asked …

.. multiple changes in the physiology and the cognitive skills ..

-> .. multiple changes in physiology and cognitive skills ..

..(e.g. in coda position

..(e.g., in coda position …

(i.e., always place a comma after each i.e., and e.g.,…)

..perceptual judgement experiment

->..perceptual judgment experiment

“and might be more common languages with relatively simple syllable structures such as Hebrew and Spanish ” - this is not grammatical; i.e., no verb

..consists in using ASR systems

-> consists of using ASR systems

..databases it should be possible explore phonetic trends in the social network

->..databases it should be possible to explore phonetic trends in the social network

..is not be possible today

-> ..is not possible today

consists in recognizing

-> .. consists of recognizing

(make this change throughout…)

“..standard (i.e. optimal) configuration of most ASR systems is characterized by a poor temporal resolution ..” - While short windows are indeed standard, why would you state that that is both optimal AND poor?

“..a loss of those acoustic events ..” - how do 10-ms frames cause such a loss? also, which events? Also, citing 18 makes no sense here, as that is merely the Kaldi tool, which does not state research results.

“ASR systems may not be able to detect the difference between poorly articulated and a clearly articulated speech sounds” - of course not, they are not designed to do that; ASR translates speech to text, ASR does not attempt such other discriminations

not be sufficientely severe to be

-> not be sufficiently severe to be

…they might be detected by an ASR system.. - this paper seems to have a misconception of ASR tasks; ASR does not detect problems with speech signals

if the databased used to

-> if the database used to

…anticipated that the system would find differences …- there is a huge difference between “the system” discovering ideas and researchers using a system to find out things. One can indeed use ASR on different datasets and draw conclusions on differences in results; however, the ASR is not designed to locate such differences

25 ms. overlapping

-> 25 ms overlapping

(do not put a period after ms)

..to maximize the error lists … -why do that?

composed by a total of

-> composed of a total of

annonimized -> anonymized

960 different stimuli (2 takers

-> 960 different stimuli (2 talkers

240 tokens: 1 takers x

-> 240 tokens: 1 talkers x

the individual speaekers, specially

-> the individual speakers, especially

…three times more frequent in the MA group (4.8%) than in YA group (2.0%), -check the math

miss-produced relatively

-> mis-produced relatively

Fig 1 -> Fig. 1 (same for all cases of “Fig”)

duration equal of below .45 seconds.

-> duration equal or below .45 seconds.

male and female speaker we

-> male and female speakers we

would have not effect

-> would have no effect

A close inspections of

-> A close inspection of

ration of consonant errors

-> ratios of consonant errors

unrelated with the actual

-> unrelated to the actual

how the ASR system recognize them.

-> how the ASR system recognizes them.

…(70% versus 70%), … -this is not “above”

resulst of the ASR system

-> results of the ASR system

of a speech in noise recognition task.

-> of speech in a noise recognition task.

..have some striking coincidences: … - hardly surprising, as both emulate human audition

“The temporal resolution is limited in ASR systems as the ones used in this study because it uses 25 ms window frames and 10 ms steps;” - do not state this as a fixed limitation; it is simple to modify window and step size in ASR; choices of 25 and 10 are very empirical, to balance accuracy and cost

..reason why the ASR system… are related ..

-> .. reason why the ASR system… is related ..

“…reason why the ASR system consistently found relatively low scores for this feature in the MA group are related with its technical limitations …” - to pursue this line of argument would require more detailed analysis, rather then pure speculation.

technically limitations of the ASR system,

-> technical limitations of the ASR system,

“within speaker differences for Ws and NWs, within utterance prosodic positions differences and also difference ..”

-> “within-speaker differences for Ws and NWs, within-utterance prosodic positions differences and also differences..”

“..[29] … not the presence or absence of periodicity that serves to recognize it;” - there are many papers in the literature for stop voicing that note a wide range of acoustic features for this contrast, and not just “tension”

“…that while the ASR system humans might be similar …” -bad grammar here; fix error

(the text repeatedly says “related with”; replace all with “related to”)

“ASR system provides a reliable approximation to articulation accuracy ..” - no, ASR just outputs a text interpretation; it does not do this

…or, alternative, with minor physiological changes.

-> …or, alternatively, with minor physiological changes.

..the difference between YA and MA speakers reflect

-> ..the difference between YA and MA speakers reflects

has fewer member than

-> …has fewer members than

…groups of sounds than might

->… groups of sounds that might

words using exclusively of vowels

-> …words using exclusively vowels

consecuence of the increased

-> consequence of the increased

vowel accuracy becasue the former

-> vowel accuracy because the former

To conclude, ther results of

-> To conclude, the results of

in everyday communcation);

-> in everyday communication);

3) intelligiblity depends on

-> 3) intelligibility depends on

Reviewer #2: The paper presents a study on the degree of degradation in speech with aging. It was observed that consonant utterances suffer more than vowels. Two group of young adults and middle aged adults were considered in the study. Degradation amount was quantified by performances of ASR as well as a human listeners in a noisy background. Justification for this particular mode of evaluation was provided.

Error analysis was performed for the individual Vowels of the Spanish language bot in the context of valid words and non-word. A significant degradation was observed.

The paper is well presented. The experimental studies are exhaustive. The experiments are well designed.

I have the following review comments:

1. The observations should be justified in terms of the associated physiological mechanism of speech production. This will enab;le the use of the study for speech therapists.

2. Instead of using the ASR as a blackbox, authors may analyse the signal features utilized by the ASR and study the changes in them. This will enable use of the investigations in development of better ASR systems.

3. Besides age, subjects with other physiological conditions and from varying dialects in Spanish should be included in the study.

6. PLOS authors have the option to publish the peer review history of their article (what does this mean?). If published, this will include your full peer review and any attached files.

Reviewer #1: No

Reviewer #2: **Yes: **Pabitra Mitra

---

## [Author Response · Author response to Decision Letter 0]

9 Oct 2020

Note: A Microsoft Word file has also been included as Response to Reviewers.

5. Review Comments to the Author

Please use the space provided to explain your answers to the questions

above. You may also include additional comments for the author, including

concerns about dual publication, research ethics, or publication ethics.

(Please upload your review as an attachment if it exceeds 20,000

characters)

Reviewer #1: Consonant and vowel articulation accuracy in younger and

middle-aged Spanish healthy adults

This study explores a hypothesis that consonants are more vulnerable to

aging than vowels. The authors used an ASR system for Spanish word (W) and

non--word (NW) repetitions. They found higher accuracy with the NWs of

younger adults than with older people, and similarly when asked to

recognize isolated syllables in background noise.

I applaud the authors for attempting to interpret their results in various

ways, but there is too much speculation here, with insufficient evidence

to support many potential explanations. Also, the authors seem unaware of

the large literature on stop voicing perception, citing only one reference

29. Yes, Spanish is different from other languages, but stop voicing

perception occurs in many languages.

RESPONSE:

1) Regarding the observation that some aspects are speculative, we have re-written several parts of the discussion, adding more specific details that support our conclusions. 

2) Regarding “stop voicing perception” literature, we have added a note about consonant confusions in other languages (English and French). We observe that in contrast with this study, other studies have provided evidence of frequent voicing errors (Meyer et al., 2013; Pathak et al, 2007). However, when the voiced/unvoiced consonant pairs which are commonly confused in those languages (in onset position) are examined in detail we observe that, for each pair both or at least one of the consonants do not exist in Spanish (e.g. ʃ/ʒ s/z). This may partly explain that errors are less common in Spanish. 

Using “Preview” on the PDF file, all of the Figures’ content was located

at the end of the file (annoying).

Particular changes needed:

1) 

…a group of healthy native listeners were asked …

-> …a group of healthy native listeners was asked …

Corrected

2) 

.. multiple changes in the physiology and the cognitive skills ..

-> .. multiple changes in physiology and cognitive skills ..

Changed

3) 

..(e.g. in coda position

..(e.g., in coda position …

(i.e., always place a comma after each i.e., and e.g.,…)

Changed

4) 

.. perceptual judgement experiment

->..perceptual judgment experiment

Changed

5) 

“and might be more common languages with relatively simple syllable

structures such as Hebrew and Spanish ” - this is not grammatical; i.e.,

no verb

Added: in

“and might be more common in languages with relatively simple syllable

structures such as Hebrew and Spanish ”

6) 

..consists in using ASR systems

-> consists of using ASR systems

After checking this, we believe it should be “in”. 

Consist of is used to describe the parts of one object

Consists in is used to describe the essential feature (generally abstract) og something. 

In this example, using the ASR is the essential feature of this methodology. 

7) 

 ..databases it should be possible explore phonetic trends in the social network

->..databases it should be possible to explore phonetic trends in the social network

Changed

8) 

..is not be possible today

-> ..is not possible today

Changed

9) 

consists in recognizing

-> .. consists of recognizing

(make this change throughout…)

As in 6, this was left unchanged

10) 

“..standard (i.e. optimal) configuration of most ASR systems is

characterized by a poor temporal resolution ..” - While short windows are

indeed standard, why would you state that that is both optimal AND poor?

“..a loss of those acoustic events ..” - how do 10-ms frames cause such a

loss? also, which events? Also, citing 18 makes no sense here, as that is

merely the Kaldi tool, which does not state research results.

“ASR systems may not be able to detect the difference between poorly

articulated and a clearly articulated speech sounds” - of course not, they

are not designed to do that; ASR translates speech to text, ASR does not

attempt such other discriminations

Regarding the first and third comment above, we acknowledge that this section was somewhat confusing. Accordingly, this section has been re-written to make to clearer. Also more relevant references have been added. . 

Original paragraph: 

Note that ideally we would expect the ASR system to recognize all and only those speech sounds that are produced accurately (according to an ideal or expert listener). However, this ideal situation \\hl{is not} possible today because Human Speech Recognition (HSR) and ASR are not identical processes. For instance, an essential aspect of speech processing in humans consists \\hl{in} recognizing very short acoustic events (e.g.\\hl{,} / p / explosion in Spanish) or rapidly changing ones (e.g.\\hl{,} formant transitions used to recognize the place of articulation of consonants and also diphthongs). Unfortunately, the standard (i.e.\\hl{,} optimal) configuration of most ASR systems is characterized by a \\hl{low} temporal resolution (i.e.\\hl{, 25 ms} window frames and 10 \\hl{ms} steps), which may result in a loss of those acoustic events (\\cite{Povey2011}). This means that, for some speech sounds, today's ASR systems may not be able to detect the difference between poorly articulated and a clearly articulated speech sounds. These considerations show that it is necessary to complement ASR data with human based analyses \\cite{Kong2017}. One such approach consists \\hl{in} asking naïve listeners to recognize speech sound presented in adverse conditions (e.g.\\hl{,} with a background noise; \\cite{Moreno-Torres2017})

This has been re-written to:

Note that ideally we would expect the ASR system to recognize all and only those speech sounds that are produced accurately (according to an ideal or expert listener. \\hl{However, previous studies comparing different ASR systems and Human Speech Recognition (HSR) have shown that, despite the overall performance level similarities, the two might not always rely on the same properties of the acoustic speech waveform (see \\mbox {\\cite{Sroka2005, Cooke2006, Meyer2007}}). These considerations show that it is necessary to complement ASR data with human based analyses \\mbox {\\cite{Kong2017}}. These considerations show that it is necessary to complement ASR data with human based analyses \\cite{Kong2017}. One such approach consists \\hl{in} asking naïve listeners to recognize speech sound presented in adverse conditions (e.g.\\hl{,} with a background noise; \\cite{Moreno-Torres2017})

11) 

not be sufficiently severe to be

-> not be sufficiently severe to be

Changed

12) 

…they might be detected by an ASR system.. - this paper seems to have a

misconception of ASR tasks; ASR does not detect problems with speech

signals

Changed to: 

they might be quantified with the help of an ASR system

13) 

if the databased used to

-> if the database used to

Changed

14) 

…anticipated that the system would find differences …- there is a huge

difference between “the system” discovering ideas and researchers using a

system to find out things. One can indeed use ASR on different datasets

and draw conclusions on differences in results; however, the ASR is not

designed to locate such differences

Original paragraph: 

anticipated that the system would find differences between the two groups, particularly for consonants, which would confirm our hypothesis.

Changed to: 

anticipated that ASR system would be more successful in recognizing the consonants of the YA than those of MA speakers, which would confirm our hypothesis.

15)

25 ms. overlapping

-> 25 ms overlapping

(do not put a period after ms)

Changed

16)

..to maximize the error lists … -why do that?

The analysis consists basically in comparing the number of errors produced by each group. One approach would consist in using one part of the corpus to train the system and another part for evaluation. Using a cross-evaluation increases the amount of data (i.e. the number of errors). 

17)

composed by a total of

-> composed of a total of

Changed

18)

annonimized -> anonymized

Changed

19)

960 different stimuli (2 takers

-> 960 different stimuli (2 talkers

240 tokens: 1 takers x

-> 240 tokens: 1 talkers x

Changed

20)

the individual speaekers, specially

-> the individual speakers, especially

Changed

21)

…three times more frequent in the MA group (4.8%) than in YA group (2.0%),

-check the math

Changed to:

between two and three times

22)

miss-produced relatively

-> mis-produced relatively

Changed

23)

Fig 1 -> Fig. 1 (same for all cases of “Fig”)

Changed

24)

duration equal of below .45 seconds.

-> duration equal or below .45 seconds.

Changed

25)

male and female speaker we

-> male and female speakers we

Changed

26)

would have not effect

-> would have no effect

A close inspections of

-> A close inspection of

ration of consonant errors

-> ratios of consonant errors

unrelated with the actual

-> unrelated to the actual

how the ASR system recognize them.

-> how the ASR system recognizes them.

All changed

27)

…(70% versus 70%), … -this is not “above”

Above changed to:

identical 

28)

resulst of the ASR system

-> results of the ASR system

of a speech in noise recognition task.

-> of speech in a noise recognition task.

All changed

29)

..have some striking coincidences: … - hardly surprising, as both emulate

human audition

Changed to: 

Are similar in various aspects.

30) 

“The temporal resolution is limited in ASR systems as the ones used in

this study because it uses 25 ms window frames and 10 ms steps;” - do not

state this as a fixed limitation; it is simple to modify window and step

size in ASR; choices of 25 and 10 are very empirical, to balance accuracy

and cost

..reason why the ASR system… are related ..

-> .. reason why the ASR system… is related ..

“…reason why the ASR system consistently found relatively low scores for

this feature in the MA group are related with its technical limitations …”

- to pursue this line of argument would require more detailed analysis,

rather than pure speculation.

We acknowledge that this section was a bit confusing and misleading to the reader. We thank the reviewer for this observation. We have modified the whole paragraph to make it clearer. 

Previous paragraph was: 

\\subsection*{ASR as model of Human Speech Recognition}

One first consideration when discussing the reliability of ASR data is whether or not the technical limitations of standard ASR systems, particularly related to the limited temporal resolution, may bias the results. 

In order to understand the phonological consequences (i.e.\\hl{,} expected error types) produced by standard ASR systems, it can be of help to consider the case of Cochlear Implant (CI) users, as CI devices and ASR systems signal processing \\hl{are similar in various aspects}: in both cases a relatively small number of spectral bands is used, and in both cases temporal resolution is relatively small \\cite{Do2010}. 

The temporal resolution is limited in ASR systems as the ones used in this study because it uses 25 ms window frames and 10 ms steps; in CI users the limitation seems to arise from the difficulty to recognize rapid changes in the electric signal sent to the auditory nerve \\cite{Bouton2012}. 

Research on CI users has consistently shown that poor temporal resolution is a major limitation of CIs and it results in certain phonological errors; considering only consonants and vowels, the most commonly cited errors are those of place or articulation (see \\cite{Bouton2012,Moreno-Torres2014}). 

Thus, it might be argued that the reason why the ASR system consistently found relatively low scores for this feature in the MA group \\hl{is related to} its technical limitations rather than with decline in articulation skills.

However, there are reasons to consider that this result is not merely a statistical 378

artifact associated with the ASR system. Note that place of articulation errors were the 379

most common ones observed during the transcription process, and these errors were 380

more frequent in the MA than in the YA group. Similarly, in the speech-in-noise task 381

the judges produced more place of articulation errors than other types, and the former 382

were more frequent with the MA speaker than with the YA speaker. Finally, it seems 383

reasonable that dynamic acoustic cues and other short acoustic events (e.g. formant 384

transitions and stop explosions) are more vulnerable than other cues (e.g. formants in 385

stressed vowels) to a decline in cognitive skills (e.g. motor sequencing) or to 386

physiological changes in the articulatory system. Thus we conclude that the reason why 387

the system found more errors of place of articulation is not only due to the technically 388

limitations of the ASR system, but also to the intrinsic vulnerability of this feature.

Changed to:

\\subsection*{ASR as model of Human Speech Recognition}

\\hl{As noted in the introduction, it is unclear to what extent ASR systems such as the one used here access the same acoustic information in the speech signal as humans do. A detailed analysis of this issue is out of the scope of this study. However, given that the results indicate that the place of articulation feature provides a key contrast between the YA and the MA speakers, and given that this feature was the worst recognized one by the ASR system, we will briefly consider the results for this feature. Note that a primary cue to the place of articulation are formant transitions \\mbox {\\cite {Green1997}}, and that recognizing formant transitions requires access to fine temporal fine information (see \\mbox{\\cite{Bouton2012}}). As in this study we used a 25 ms window, it is possible that the ASR recognized poorly the formant transitions.} This means that a decrease is the rate of recognition as observed in the results might not necessarily imply that a decline in articulation skills does take place. However, there are reasons to consider that the results of this study, at least as regards the differences between YA and MA speakers, are not merely a statistical artifact. 

\\hl{Note that the place of articulation errors were the most common ones observed during the transcription process, and that these errors were more frequent in the MA group than in the YA group. Two possible causes may explain these errors. One is that the MA group had minor auditory temporal processing deficits \\mbox{\\cite{Pichora2003, helfer_vargo_2009}}, which may lead to poor recognition of the place of articulation \\mbox{\\cite{Bouton2012}} and which, given the close link between perception and articulation, may negatively impact articulation accuracy selectively for this feature \\mbox{\\cite{tourville2011}}. Alternatively, it might occur that the errors observed by the human experts were due to a decline in cognitive skills supporting articulation (e.g., motor sequencing \\mbox{|\\cite{Bilodeau-Mercure2016}}). Thus, independently of the underlying cause, the articulation skills of the MA group might be poorer than those of the YA group, which agrees with the data obtained with the ASR system.}

32) 

technically limitations of the ASR system,

-> technical limitations of the ASR system,

This has been removed

33) 

“within speaker differences for Ws and NWs, within utterance prosodic

positions differences and also difference ..”

-> “within-speaker differences for Ws and NWs, within-utterance prosodic

positions differences and also differences..”

changed

34) 

“..[29] … not the presence or absence of periodicity that serves to

recognize it;” - there are many papers in the literature for stop voicing

that note a wide range of acoustic features for this contrast, and not

just “tension”

Please note that “tension” is not a single acoustic cue. Rather it is the underlying cause that modifies multiple acoustic cues. As noted, in the text, cues affected by tension include F0, the vowel formants, intensity, etc. However, tension need not be equally relevant in all languages. Here we merely adopt Martinez Celdrán proposal for Spanish.

35) 

“…that while the ASR system humans might be similar …” -bad grammar here;

fix error

(the text repeatedly says “related with”; replace all with “related to”)

changed

36) 

“ASR system provides a reliable approximation to articulation accuracy ..”

- no, ASR just outputs a text interpretation; it does not do this

Original paragraph: 

To conclude, our results indicate that there are many aspects for which the ASR system provides a reliable approximation to articulation accuracy, and this includes gross categories such as consonants and vowels and also consonant features. However, it is unclear to what extent the results for specific consonants match the results in humans.

Modified to: 

\\hl{To conclude, the analysis of the results shows that the data obtained with the ASR experiment served to compute reliable measures of articulation accuracy for gross categories such as consonants and vowels, and also consonant features. However, it is unclear to what extent the measures obtained for specific consonants provide a reliable approximation to the accuracy observed in humans.}

37) 

…or, alternative, with minor physiological changes.

-> …or, alternatively, with minor physiological changes.

..the difference between YA and MA speakers reflect

-> ..the difference between YA and MA speakers reflects

has fewer member than

-> …has fewer members than

…groups of sounds than might

->… groups of sounds that might

words using exclusively of vowels

-> …words using exclusively vowels

consecuence of the increased

-> consequence of the increased

vowel accuracy becasue the former

-> vowel accuracy because the former

To conclude, ther results of

-> To conclude, the results of

in everyday communcation);

-> in everyday communication);

3) intelligiblity depends on

-> 3) intelligibility depends on

All changed

Reviewer #2: The paper presents a study on the degree of degradation in

speech with aging. It was observed that consonant utterances suffer more

than vowels. Two group of young adults and middle aged adults were

considered in the study. Degradation amount was quantified by performances

of ASR as well as a human listeners in a noisy background. Justification

for this particular mode of evaluation was provided.

Error analysis was performed for the individual Vowels of the Spanish

language bot in the context of valid words and non-word. A significant

degradation was observed.

The paper is well presented. The experimental studies are exhaustive. The

experiments are well designed.

I have the following review comments:

1. The observations should be justified in terms of the associated

physiological mechanism of speech production. This will enable the use of

the study for speech therapists.

We thank the reviewer for this observation. In the present version analyze the possibility that articulatory complexity and / perceptual factors may contribute to a decay in consonant articulation. To address this, changes have been added: 

- In the introduction: 

\\hl{Also, there is increasing evidence that auditory processing skills tend to decay with age \\mbox{\\cite{helfer_vargo_2009,Recanzone2017}}, and that this may impact speech articulation skills\\mbox{\\cite{tourville2011}}}.

- In the results

However, in the case of the female participants there was a weak but significant correlation between the consonants and place of articulation errors annotated by the human experts and the corresponding measures computed from the ASR results. For consonants: Spearman r = .39; s = .021. For the place of articulation feature: r = .34; s = .047). This indicates that in the case of the female participants there might be a link between the two measures. }

- In the discussion

\\hl{Finally, it seems relevant to consider to what extent auditory perception may contribute to these results. Note that recent neurolinguistic models of speech production have shown that the speech production system includes auditory and somatosensory feedback mechanisms that are used to control articulation accuracy \\mbox{\\cite{tourville2011}}. It is also relevant that there are age-related changes in speech perception (i.e., peripheral high-frequency hearing loss and central deficits of auditory temporal processing; see \\mbox{\\cite{Pichora2003, helfer_vargo_2009}}). And also that the recognition of the place of articulation feature requires efficient temporal auditory processing skills. Thus it is possible that minor deficits in auditory temporal processing result in impoverished feedback and, in the end, contribute to a reduction of the accuracy in place of articulation feature. Note that this link (though not the casual relationship) was confirmed only in the case of the female participants, a result that is compatible with the evidence that that there are sex-related differences in the development and decay of these skills \\mbox{\\cite{helfer_vargo_2009}}. }

2. Instead of using the ASR as a blackbox, authors may analyse the signal

features utilized by the ASR and study the changes in them. This will

enable use of the investigations in development of better ASR systems.

We acknowledge that a detailed study of the signal features used by the ASR system

and inherent changes is an interesting topic and would be beneficial to improve ASR

technology. In our paper, our main goal is to get a better understanding of a

linguistic problem, so our set of ASR parameters have been chosen to mimic/replicate

human performance as better as possible. We did some simulations with different

feature parameters (window size, overlapping strategy, feature selection). The results revealed that some differences, which were not significant for any of the phonetic aspects analyzed in this study. Thus, we

choose to use a unique set of parameters (with good replication of human

performance) to avoid the effect of too many variables in our study. 

3. Besides age, subjects with other physiological conditions and from

varying dialects in Spanish should be included in the study.

We agree with the reviewer. However, the large variability of speech articulation (register, dialect or even the psychological state) would make it most difficult to create a balanced corpus, which was necessary to answer this specific research questions addressed in this study. Hopefully, in the near future we will be able to use the same methodology for larger and more diverse groups of speakers, as suggested by the reviewer.

---

## [Decision Letter · Decision Letter 1]

21 Oct 2020

PONE-D-20-18733R1

Consonant and vowel articulation accuracy in younger and middle-aged Spanish healthy adults

PLOS ONE

Dear Dr. Moreno Torres,

Thank you for submitting your manuscript to PLOS ONE. After careful consideration, we feel that it has merit but does not fully meet PLOS ONE’s publication criteria as it currently stands. Therefore, we invite you to submit a revised version of the manuscript that addresses the points raised during the review process.

**Dear Dr. Moreno Torres,**

**We are interested in publishing your submission “Consonant and vowel articulation accuracy in younger and middle-aged Spanish healthy adults” in PLOS ONE.  The manuscript does, however, require some further revision. We ask that you consider revising with respect to the changes suggested. **

** **

**Specifically, a few minor changes are given by Reviewer 1. In addition, your earlier response to Reviewer 2 needs to be included in the main text (see below).**

** Your earlier response: We agree with the reviewer. However, the large variability of speech articulation (register, dialect or even the psychological state) would make it most difficult to create a balanced corpus, which was necessary to answer this specific research questions addressed in this study. Hopefully, in the near future we will be able to use the same methodology for larger and more diverse groups of speakers, as suggested by the reviewer **

**-> The above information should be mentioned specifically as a potential limitation and/or direction of further extension of the present investigation**

**Please ensure you display the changes to your revised manuscript by using either the highlighter function in MS Word, or by using bold, underlined, or colored text. This will greatly help peer reviewers evaluate your revised submission. When submitting your new revision, a point-by-point response to the comments by Reviewer 1 and myself is optional. **

We look forward to receiving your revised manuscript.

Kind regards,

Anthony Pak-Hin Kong, Ph.D.

Academic Editor

PLOS ONE

Additional Editor Comments (if provided):

Dear Dr. Moreno Torres,  

We are interested in publishing your submission “Consonant and vowel articulation accuracy in younger and middle-aged Spanish healthy adults” in PLOS ONE.  The manuscript does, however, require some further revision. We ask that you consider revising with respect to the changes suggested. 

Specifically, a few minor changes are given by Reviewer 1. In addition, your earlier response to Reviewer 2 needs to be included in the main text (see below).

Your earlier response: We agree with the reviewer. However, the large variability of speech articulation (register, dialect or even the psychological state) would make it most difficult to create a balanced corpus, which was necessary to answer this specific research questions addressed in this study. Hopefully, in the near future we will be able to use the same methodology for larger and more diverse groups of speakers, as suggested by the reviewer

-> The above information should be mentioned specifically as a potential limitation and/or direction of further extension of the present investigation  

Please ensure you display the changes to your revised manuscript by using either the highlighter function in MS Word, or by using bold, underlined, or colored text. This will greatly help peer reviewers evaluate your revised submission. When submitting your new revision, a point-by-point response to the comments by Reviewer 1 and myself is optional.

Reviewers' comments:

Reviewer's Responses to Questions

**Comments to the Author**

1. If the authors have adequately addressed your comments raised in a previous round of review and you feel that this manuscript is now acceptable for publication, you may indicate that here to bypass the “Comments to the Author” section, enter your conflict of interest statement in the “Confidential to Editor” section, and submit your "Accept" recommendation.

Reviewer #1: All comments have been addressed

2. Is the manuscript technically sound, and do the data support the conclusions?

Reviewer #1: Yes

3. Has the statistical analysis been performed appropriately and rigorously? 

Reviewer #1: Yes

4. Have the authors made all data underlying the findings in their manuscript fully available?

Reviewer #1: Yes

5. Is the manuscript presented in an intelligible fashion and written in standard English?

Reviewer #1: Yes

6. Review Comments to the Author

Reviewer #1: a few minor changes needed:

…identical in both group of speakers …->

…identical in both groups of speakers …

Note that a primary cue to the place of articulation are …

-> Note that primary cues to the place of articulation are …

..means that a decrease is the rate of recognition… ->

..means that a decrease in the rate of recognition…

..in Spanish language. ->

..in the Spanish language.

..versus vowels) the finer the categories…

-> …versus vowels); the finer the categories..

..in one condition exclusively). >

..in one condition exclusively.

..part of these errors are caused by ..->

..some of these errors are caused by ..

7. PLOS authors have the option to publish the peer review history of their article (what does this mean?). If published, this will include your full peer review and any attached files.

Reviewer #1: No

---

## [Author Response · Author response to Decision Letter 1]

23 Oct 2020

Dear Dr Kong,

Thank you for your letter. We have included all minor changes suggested by Reviewer 1. Also, as suggested by reviewer 2, a paragraph noting some limitations of this study (i.e. regarding the need to consider language variation) has been added in the Conclusions (final section) of the manuscript.

Best regards,

---

## [Editor Report · Decision Letter 2]

26 Oct 2020

Consonant and vowel articulation accuracy in younger and middle-aged Spanish healthy adults

PONE-D-20-18733R2

Dear Dr. Moreno Torres,

We’re pleased to inform you that your manuscript has been judged scientifically suitable for publication and will be formally accepted for publication once it meets all outstanding technical requirements.

Kind regards,

Anthony Pak-Hin Kong, Ph.D.

Academic Editor

PLOS ONE

Additional Editor Comments (optional):

Dear Dr. Moreno Torres,

  I am pleased to accept your manuscript PONE-D-20-18733R2 “Consonant and vowel articulation accuracy in younger and middle-aged Spanish healthy adults” for publication in PLOS ONE.  

Thank you for the opportunity to review and publish your work. 

Sincerely,  

Anthony Pak-Hin Kong, Ph.D.

 Academic Editor 

PLOS ONE
---

## [Editor Report · Acceptance letter]

30 Oct 2020

PONE-D-20-18733R2 

Consonant and vowel articulation accuracy in younger and middle-aged Spanish healthy adults 

Dear Dr. Moreno Torres:

I'm pleased to inform you that your manuscript has been deemed suitable for publication in PLOS ONE. Congratulations! Your manuscript is now with our production department. 

Kind regards, 

on behalf of

Dr. Anthony Pak-Hin Kong 

Academic Editor

PLOS ONE